# Biological sample donation and informed consent for neurobiobanking: Evidence from a community survey in Ghana and Nigeria

Arti Singh[1], Oyedunni Arulogun[2], Joshua Akinyemi[3], Michelle Nichols[4], Benedict Calys-Tagoe[5], Babatunde Ojebuyi[6], Carolyn Jenkins[4], Reginald Obiako[7], Albert Akpalu[5], Fred Sarfo[8], Kolawole Wahab[9], Adeniyi Sunday[9], Lukman F. Owolabi[10], Muyiwa Adigun[11], Ibukun Afolami[12], Olorunyomi Olorunsogbon[13], Mayowa Ogunronbi[13], Ezinne Sylvia Melikam[14], Ruth Laryea[5], Shadrack Asibey[8], Wisdom Oguike[7], Lois Melikam[7], Abdullateef Sule[7], Musibau A. Titiloye[2], Isah Suleiman Yahaya[10], Abiodun Bello[9], Rajesh N. Kalaria[15], Ayodele Jegede[16], Mayowa Owolabi[14], Bruce Ovbiagele[17], Rufus Akinyemi[13]*

1 School of Public Health, Kwame Nkrumah University of Science and Technology, Kumasi, Ghana, 2 Department of Health Promotion and Education, Faculty of Public Health, University of Ibadan, Ibadan, Nigeria, 3 Department of Epidemiology and Medical Statistics, College of Medicine, University of Ibadan, Ibadan, Nigeria, 4 College of Nursing, Medical University of South Carolina, Carolina, Charleston, United Sates of America, 5 University of Ghana Medical School, College of Health Sciences, Accra, Ghana, 6 Department of Communication and Language Arts, Faculty of Arts, University of Ibadan, Nigeria, 7 Neurology Unit, Department of Medicine, Ahmadu Bello University Teaching Hospital, Shika, Zaria, Nigeria, 8 Neurology Unit, Department of Medicine, Kwame Nkrumah University of Science and Technology, Kumasi, Ghana, 9 Neurology Unit, Department of Medicine, University of Ilorin Teaching Hospital, University of Ilorin, Ilorin, Nigeria, 10 Neurology Unit, Department of Medicine, Aminu Kano Teaching Hospital, Bayero University, Kano, Nigeria, 11 Faculty of Law, University of Ibadan, Ibadan, Nigeria, 12 Faculty of Public Health, College of Medicine, University of Ibadan, Ibadan, Nigeria, 13 Neuroscience and Ageing Research Unit, Institute for Advanced Medical Research and Training, College of Medicine, University of Ibadan, Ibadan, Nigeria, 14 Department of Medicine, College of Medicine, University of Ibadan, Ibadan, Nigeria, 15 Neurovascular Research Group, Institute of Neuroscience, Newcastle University, Newcastle upon Tyne, United Kingdom, 16 Department of Sociology, Faculty of the Social Sciences, University of Ibadan, Ibadan, Nigeria, 17 School of Medicine, University of California, San Francisco, San Francisco, California, United States of America

* artisingh_uk@yahoo.com

## Abstract

### Introduction

Genomic research and neurobiobanking are expanding globally. Empirical evidence on the level of awareness and willingness to donate/share biological samples towards the expansion of neurobiobanking in sub-Saharan Africa is lacking.

### Aims

To ascertain the awareness, perspectives and predictors regarding biological sample donation, sharing and informed consent preferences among community members in Ghana and Nigeria.

### Methods

A questionnaire cross-sectional survey was conducted among randomly selected community members from seven communities in Ghana and Nigeria.

**Data Availability Statement:** All relevant data are within the paper and its Supporting Information files.

**Funding:** Our work is supported by The National Institutes of Health grants: African Neurobiobank for Precision Stroke Medicine ELSI Project (U01HG010273), SIREN (U54HG007479), SIBS Genomics (R01NS107900), SIBS Gen Gen (R01NS107900?02S1), ARISES (R01NS115944?01), H3Africa CVD Supplement (3U24HG009780?03S5), CaNVAS (1R01NS114045-01), Sub-Saharan Africa Conference on Stroke (SSACS) 1R13NS115395-01A1 and Training Africans to Lead and Execute Neurological Trials & Studies (TALENTS) D43TW01203.

**Competing interests:** The authors have declared that no competing interests exist.

## Results

Of the 1015 respondents with mean age 39.3 years (SD 19.5), about a third had heard of blood donation (37.2%, M: 42.4%, F: 32.0%, p = 0.001) and a quarter were aware of blood sample storage for research (24.5%; M: 29.7%, F: 19.4%, p = 0.151). Two out of ten were willing to donate brain after death (18.8%, M: 22.6%, F: 15.0%, p<0.001). Main reasons for unwillingness to donate brain were; to go back to God complete (46.6%) and lack of knowledge related to brain donation (32.7%). Only a third of the participants were aware of informed consent (31.7%; M: 35.9%, F: 27.5%, p<0.001). Predictors of positive attitude towards biobanking and informed consent were being married, tertiary level education, student status, and belonging to select ethnic groups.

## Conclusion

There is a greater need for research attention in the area of brain banking and informed consent. Improved context-sensitive public education on neurobiobanking and informed consent, in line with the sociocultural diversities, is recommended within the African sub region.

## Introduction

Biobanking and genomic research are becoming increasingly important for health and disease research in developing countries including the African sub region. There are increasing efforts to capture global genetic diversity in an attempt to ensure that the benefits of genomic innovation filter down to all people around the globe [1]. Neurobiobanking, the storage of central nervous system tissues, including fixed and frozen whole brain, brain sections, brain biopsies, spinal cord, associated blood fractions, and relevant datasets stored for research purposes, is also expanding in Africa [2]. With the huge human genomic diversity, coupled with an ageing population and associated brain disorders, the Ibadan Brain Ageing, Dementia And Neurodegeneration (IBADAN) Brain Bank [2], the first organized brain tissue biorepository in sub-Saharan Africa (sSA), was set up to accrue, process and store unique brain tissues for future research into a broad spectrum of neurological disorders such as stroke and dementias. Future discoveries emanating from these resources and systems have an immeasurable potential health benefit to people of African ancestry and other ancestral populations [3]. Despite these groundbreaking advancements in genomic research within the African research context, several questions related to the ethical, legal and social aspects of neurobiobanking remain unanswered. For instance, among Africans, communal informed consent is preferred to individual informed consent, given that the African context tends to prioritize values like communitarianism and reciprocity over respect for autonomy [4]. Also, relationship between people and considerations of community benefit are considered equally important [5]. However, the success of biobanking depends on people's willingness to contribute their biological samples for storage towards research. Public support is thus essential in securing the sustainability of biobanks. A review of studies conducted globally indicated willingness to donate by individuals, despite poor knowledge [6]. However, some studies have indicated that biobanking-knowledge, type of donated tissue, purpose of research, safety of the data, preferred type of consent, and trust towards biobanks are all influential factors related to willingness to donate [7–9]. Studies conducted in the African sub region such as Nigeria indicates a high level of awareness but poor willingness towards organ donation such as brain, among older Nigerians [10].

Nevertheless, studies conducted in Europe and America indicate a generally positive attitude towards biobanking and a high willingness to donate, but these concepts have not been substantially investigated within the African sub region [11–13].

Given the unique socio-cultural, linguistic and belief systems of Africans, the ethical, legal and social implications (ELSI) of emerging biobanks including neurobiobanks and data resources in Africa require detailed exploration [14]. The objectives of this study were to evaluate the awareness and willingness of community members in Ghana and Nigeria towards donation/sharing of blood and brain samples for neurobiobanking and genetic research and their preferences regarding informed consent for participation.

## Materials and methods

### Study design, participants, sampling and setting

A cross-sectional survey was conducted among community-dwelling laypersons from seven sites within our existing established SIREN (Stroke Investigative Research and Education Network) [1], from Ghana and Nigeria. The SIREN study has a cohort of stroke survivors, caregivers and healthy controls [15]. Participants from five (5) communities in Nigeria (Abeokuta, Ibadan, Ilorin, Kano and Zaria) and two (2) communities in Ghana (Accra and Kumasi) were included. As described in the protocol manuscript of the study community-dwelling laypersons were recruited from the seven SIREN participating sites during community engagement programmes regularly organized to promote stroke awareness in the community and also recruit suitable controls for the ongoing study [14]. The number of participants surveyed per site was proportionate to the size of SIREN recruitment from each site. Respondents were selected by stratified random sampling using the list of participants at the community engagement programme list as sampling frame. Detailed information on the seven participating sites has been published elsewhere [14]. The sample size was estimated based on preliminary data, which showed 48.7% of stroke patients and 57.0% of stroke-free individuals have knowledge of stroke heritability [16]. We used knowledge of stroke heritability as a proxy for awareness about stroke genomics research, which was one of the goals of the ELSI project. The effective sample size estimated was 975 based on a 3% degree of precision and 95% confidence level and adjusting for 85% anticipated response rate.

### Study tool and data collection

A survey questionnaire (S1 File) was used to collect information from the participants. The study questionnaire was developed by a multidisciplinary expert-working group and informed by a systematic review of the literature and our findings from previous studies [1, 2, 17].

### Survey measures

1. Information on participant demographics includes age, religion, ethnicity, educational qualification, occupation, marital status, monthly income and living arrangement.

2. Awareness and knowledge related to blood/brain fraction donation

3. Willingness to give consent for blood and brain sample donation for genetic research and storage.

4. Awareness and perception of informed consent.

Trained interviewers at each SIREN site administered the questionnaire and written informed consent was obtained prior to completion of the survey. All participants received a brief

education on the concepts being explored in the questionnaire such as informed consent and their meaning and in some cases an explanatory note was provided with the question eg. broad consent (Informed consent only needs to be taken once and this covers for all other use of my sample by researchers) for research on the blood collected from me as it is sometimes practically difficult to re-contact and re-consent participants.

### Ethical consideration

Ethical approval was sought from the Institutional Health Research review board of each participating SIREN site (the University of Ibadan; Federal Medical Centre, Abeokuta; University of Ilorin; Aminu Kano Teaching Hospital, Kano; Ahmadu Bello University, Zaria; University of Ghana, Accra and Kwame Nkrumah University of Science and Technology, Kumasi). Confidentiality of data was ensured throughout all phases of the study. Data were analyzed anonymously, with only members of the study in charge of data analysis having access to collected data. Confidentiality of data continued until the full manuscript was finalized. After publication, the data will be safely stored with continued maintenance of confidentiality.

### Statistical analysis

Descriptive statistics was used to summarize the demographic characteristics of the participants. For hypothesis testing, $\chi^2$ or Fisher's exact test was used to investigate factors associated with awareness of brain/blood donation, willingness to share blood/brain tissues with other researchers and awareness and perception of informed consent. The Mann-Whitney U test was used to analyze ranked responses including participants' preferences for receiving genetic test results. Random-effect logistic regression models were fitted to identify the socio-demographic characteristics associated with willingness to share blood/brain fraction and awareness of informed consent. For all statistical analyses, a p-value $< 0.05$ was considered significant at 95% confidence levels.

## Results

### Socio-demographic characteristics of participants

A total of 1015 community members, mean age 39.3 years (SD 19.5) with an equal representation of males and females completed the interviewer-administered questionnaire in the selected study sites in Ghana and Nigeria. The socio-demographic characteristics of the study participants are summarized in Table 1.

Table 1 shows that over 70% of the participants were below 50 years of age. Over half of the participants were married (54%) and had tertiary education (51%). Two-thirds of the participants were Christians (62.9%) and close to half of the participants belonged to the Yoruba ethnic groups (47.9%) followed by the Hausa groups (15.6%) (p< 0.001).

### Awareness of blood donation

Table 2 summarizes awareness of respondents to blood donation. Only a third of the participants had previously heard of blood donation for research (M: 42.4%, F: 32.0%, p,<0.01) and less than a third (24.5%; M: 29.7%, F: 19.4%, (p<0.01) were aware of blood sample storage for research.

As shown in Table 2, hospitals were the main source of where participants obtained information on blood donation (65.3%; M: 58.3, F: 74.7%, p<0.001) followed by internet and online sources (17.7%; M: 23.2%, F: 10.5%, p<0.001). Close to 90% (n = 913) of participants were not aware of any guidelines regulating blood sample storage for genomic research.

**Table 1. Socio-demographic characteristics of study participants.**

| Characteristics | Male (n = 509) | Female (n = 506) | Total (n = 1015) | p-value |
|---|---|---|---|---|
| Age: Mean(SD) | 39.62 (23.06) | 38.91 (15.16) | 39.27 (19.52) | 0.561 |
| Age group | n (%) | n (%) | n (%) | |
| < 50 | 387 (76.33) | 377 (74.80) | 764 (75.57) | 0.571 |
| > = 50 | 120 (23.67) | 127 (25.20) | 247 (24.43 | |
| Domicile | | | | |
| Rural | 21 (4.14) | 27 (5.36) | 48 (4.75) | 0.470 |
| Semi-urban | 131 (25.84) | 140 (27.78) | 271 (26.81) | |
| Urban (Ref) | 355 (70.02) | 337 (66.87) | 692 (68.45) | |
| Education | | | | |
| None | 16 (3.14) | 50 (9.88) | 66 (6.50) | <0.001 |
| Arabic* | 4 (0.79) | 11 (2.17) | 15 (1.48) | |
| Primary | 45 (8.84) | 74 (14.62) | 119 (11.72) | |
| Secondary | 132 (25.93) | 163 (32.21) | 295 (29.06) | |
| Tertiary | 312 (61.30) | 208 (41.11) | 520 (51.23) | |
| Average monthly income | | | | |
| < = 100 USD | 269 (54.79) | 292 (61.47) | 561 (58.07) | 0.004 |
| > 100 USD | 222 (45.21) | 183 (38.53) | 405 (41.93) | |
| Marital status | | | | |
| Single | 230 (45.19) | 160 (31.62) | 390 (38.42) | <0.001 |
| Married | 261 (51.28) | 291 (57.51) | 552 (54.38) | |
| Formerly married | 18 (3.54) | 55 (10.87) | 73 (7.19) | |
| Living arrangement | | | | |
| Alone | 160 (31.43) | 89 (17.59) | 249 (24.53) | <0.001 |
| With spouse and children | 251 (49.31) | 281 (55.53) | 532 (52.41) | |
| With children | 11 (2.16) | 52 (10.28) | 63 (6.21) | |
| With others | 87 (17.09) | 84 (16.60) | 171 (16.85) | |
| Religion | | | | |
| Christianity | 308 (60.51) | 331 (65.42) | 639 (62.96) | 0.091 |
| Islam | 196 (38.51) | 174 (34.39) | 370 (36.45) | |
| Others | 5 (0.98) | 1 (0.20) | 6 (0.59) | |
| Ethnic group | | | | |
| Yoruba | 229 (44.99) | 257 (50.79) | 486 (47.88) | <0.001 |
| Igbo | 23 (4.52) | 11 (2.17) | 34 (3.35) | |
| Hausa | 92 (18.07) | 66 (13.04) | 158 (15.57) | |
| Akan | 60 (11.79) | 84 (16.60) | 144 (14.19) | |
| Ga/Adangbe | 32 (6.29) | 34 (6.72) | 66 (6.50) | |
| Ewe | 8 (1.57) | 11 (2.17) | 19 (1.87) | |
| Others | 65 (12.77) | 43 (8.50) | 108 (10.64) | |
| Primary Occupation | | | | |
| Highly skilled | 69 (13.56) | 34 (6.72) | 103 (10.15) | <0.01 |
| Skilled | 126 (24.75) | 122 (24.11) | 248 (24.43) | |
| Semi-skilled | 84 (16.50) | 110 (21.74) | 194 (19.11) | |
| Manual work | 53 (10.41) | 96 (18.96) | 149 (14.68) | |
| Not working | 72 (14.15) | 84 (16.60) | 156 (15.37) | |
| Students | 105 (20.63) | 60 (11.86) | 165 (16.26) | |

*Arabic schools are common in Northern Nigeria. They do some sorts of formal education, but the focus is Quran. It is not equivalent to "No formal education".

**Table 2. Awareness and knowledge related to blood sample donation.**

| Variable/question | Male (n = 509) | Female (n = 506) | Total (n = 1015) | p-value |
|---|---|---|---|---|
| *Ever heard of blood sample donation for medical research* | n (%) | n (%) | n (%) | |
| Yes | 216 (42.44) | 162 (32.02) | 378 (37.24) | 0.001 |
| No | 293 (57.56) | 344 (67.98) | 637 (62.76) | |
| *Ever heard of blood sample storage for research purpose* | | | | |
| Yes | 151 (29.67) | 98 (19.37) | 249 (24.53) | <0.001 |
| No | 358 (70.33) | 408 (80.63) | 766 (75.47) | |
| *Sources of information about blood sample donation* | | | | |
| Hospital | 126 (58.33) | 121 (74.69) | 247 (65.34) | <0.001 |
| Training program | 21 (9.72) | 13 (8.02) | 34 (8.99) | 0.568 |
| Friend | 31 (14.35) | 11 (6.79) | 42 (11.11) | 0.021 |
| Colleague | 16 (7.41) | 7 (4.32) | 23 (6.08) | 0.214 |
| Newspaper/magazine | 18 (8.33) | 5 (3.09) | 23 (6.08) | 0.035 |
| Internet/online resources | 50 (23.15) | 17 (10.49) | 67 (17.72) | 0.001 |
| Seminar/conference/workshop | 23 (10.65) | 5 (3.09) | 28 (7.41) | 0.005 |
| TV | 29 (13.43) | 7 (4.32) | 36 (9.52) | 0.003 |
| Radio | 29 (13.43) | 10 (6.17) | 39 (10.32) | 0.022 |
| Outreach | 27 (12.50) | 13 (8.02) | 40 (10.58) | 0.162 |
| Family | 9 (4.17) | 6 (3.70) | 15 (3.97) | 0.820 |
| *Sources of information about blood sample storage for research* | | | | |
| Hospital | 86 (56.95) | 69 (70.41) | 155 (62.25) | 0.032 |
| Training program | 17 (11.26) | 10 (10.20) | 27 (10.84) | 0.794 |
| Friend | 18 (11.92) | 10 (10.20) | 28 (11.24) | 0.675 |
| Colleague | 8 (5.30) | 4 (4.08) | 12 (4.82) | 0.662 |
| Newspaper/magazine | 16 (10.60) | 5 (5.10) | 21 (8.43) | 0.127 |
| Internet/online resources | 32 (21.19) | 13 (13.27) | 45 (18.07) | 0.112 |
| Seminar/conference/workshop | 19 (12.58) | 3 (3.06) | 22 (8.84) | 0.010 |
| TV | 24 (15.89) | 6 (6.12) | 30 (12.05) | 0.021 |
| Radio | 18 (11.92) | 6 (6.12) | 24 (9.64) | 0.130 |
| Outreach | 15 (9.93) | 8 (8.16) | 23 (9.24) | 0.637 |
| Family | 9 (5.96) | 1 (1.02) | 10 (4.02) | 0.052 |
| *Awareness about guidelines/regulation for use of blood and its storage for genomic research* | | | | |
| Yes | 57 (11.20) | 45 (8.89) | 102 (10.05) | 0.222 |
| No | 452 (88.80) | 461 (91.11) | 913 (89.95) | |

## Awareness of brain donation

As shown in Table 3, awareness of brain donation was lower than blood donation (9.8% compared to 37.2% for blood donation), and over 60% of the respondents were unaware of any guidelines for brain donation.

Table 3 shows that nearly 9 out of 10 (89.0%) respondents were unaware of anyone who had agreed to brain donation and only 5.7% had previously heard about collecting and storing brain for research (M: 7.3%, F: 4.2%, p = 0.032). Likewise, less than a quarter were willing to donate brain after death (18.8%; M: 22.6%, F: 15.0%, p = 0.002). Main reasons for unwillingness to donate brain samples were: wanting to go back to God complete (46.6%; M: 39.9%, F: 52.8%, p<0.001), lack of knowledge (32.7%; M: 32.0%, F: 33.3%, p = 0.696) and distrust in the medical system (27.6%; M: 30.2%, F: 25.1%, p = 0.103). Over 80% (85.8%, M: 82.3%, F: 89.1%, p-0.003) of respondents disagreed with the statement: "people in Africa would be willing to

**Table 3.  Awareness and knowledge related to brain donation.**

| Variable/question | Male: (n = 509) | Female: (n = 506) | Total: (n = 1015) | p-value |
|---|---|---|---|---|
| *Ever heard of brain donation for research* | n (%) | n (%) | n (%) | |
| Yes | 53 (10.41) | 47 (9.29) | 100 (9.85) | 0.558 |
| No | 456 (89.59) | 459 (90.71) | 915 (90.15) | |
| *Sources of information about brain donation for research* | | | | |
| Hospital | 14 (26.42) | 21 (44.68) | 35 (35.00) | 0.056 |
| Training program | 7 (13.21) | 6 (12.77) | 13 (13.00) | 0.948 |
| Friend | 6 (11.32) | 6 (12.77) | 12 (12.00) | 0.824 |
| Colleague | 2 (3.77) | 0 (0.0) | 2 (2.00) | 0.179 |
| Newspaper/magazine | 8 (15.09) | 0 (0.0) | 8 (8.00) | 0.005 |
| Internet/online resources | 16 (30.19) | 8 (17.02) | 24 (24.00) | 0.124 |
| Seminar/conference/workshop | 3 (5.66) | 2 (4.26) | 5 (5.00) | 0.748 |
| TV | 13 (24.53) | 6 (12.77) | 19 (19.00) | 0.135 |
| Radio | 5 (9.43) | 5 (10.64) | 10 (10.00) | 0.841 |
| Outreach | 3 (5.66) | 2 (4.26) | 5 (5.00) | 0.748 |
| Family | 2 (3.77) | 1 (2.13) | 3 (3.00) | 0.630 |
| *Awareness about a brain donor* | | | | |
| Yes | 6 (11.32) | 5 (10.64) | 11 (11.00) | 0.913 |
| No | 47 (88.68) | 42 (89.36) | 89 (89.00) | |
| *Awareness about guidelines for use of brain for research* | | | | |
| Yes | 10 (27.03) | 9 (42.86) | 19 (32.76) | 0.217 |
| No | 27 (72.97) | 12 (57.14) | 39 (67.24) | |
| *Have you ever heard of the concept of collecting and storing brain for research*? | | | | |
| Yes | 37 (7.27) | 21 (4.15) | 58 (5.71) | 0.032 |
| No | 472 (92.73) | 485 (95.85) | 957 (94.29) | |
| *Willingness to donate brain after death* | | | | |
| Yes | 115 (22.59) | 76 (15.02) | 191 (18.82) | 0.002 |
| No | 394 (77.41) | 430 (84.98) | 824 (81.18) | |
| *Reasons for willingness* | | | | |
| It will advance medicine | 73 (63.48) | 41 (53.95) | 114 (59.69) | 0.189 |
| Prevent future disease | 59 (51.30) | 24 (31.58) | 83 (43.46) | 0.007 |
| Don't need brain after death | 38 (33.04) | 19 (25.00) | 57 (29.84) | 0.234 |
| Happy to safe a life | 63 (54.78) | 37 (48.68) | 100 (52.36) | 0.409 |
| It can help future generations | 41 (35.65) | 20 (26.32) | 61 (31.94) | 0.176 |
| *Reasons for not willing to donate brain sample* | | | | |
| Want to go back to God complete | 157 (39.85) | 227 (52.79) | 384 (46.60) | <0.001 |
| Against my religion | 63 (15.99) | 62 (14.42) | 125 (15.17) | 0.530 |
| People will think I am occultic | 60 (15.23) | 34 (7.91) | 94 (11.41) | 0.001 |
| Not knowledgeable about it | 126 (31.98) | 143 (33.26) | 269 (32.65) | 0.696 |
| Don't trust Africans | 18 (4.57) | 28 (6.51) | 46 (5.58) | 0.225 |
| Don't trust medical systems | 119 (30.20) | 108 (25.12) | 227 (27.55) | 0.103 |
| It's like destroying the work of God | 22 (5.58) | 24 (5.58) | 46 (5.58) | 0.999 |
| I don't just want to | 17 (4.31) | 33 (7.67) | 50 (6.07) | 0.044 |
| *People in Africa would be willing to donate brain samples for research purposes* | | | | |
| Agree | 88 (17.43) | 55 (10.91) | 143 (14.17) | 0.003 |
| Disagree | 417 (82.57) | 449 (89.09) | 866 (85.83) | |
| *Actions for promoting brain sample donation for research* | | | | |
| Media publicity | 325 (63.85) | 329 (65.02) | 654 (64.43) | 0.697 |

*(Continued)*

**Table 3.** (Continued)

| Variable/question | Male: (n = 509) | Female: (n = 506) | Total: (n = 1015) | p-value |
|---|---|---|---|---|
| Education | 332 (65.23) | 314 (62.06) | 646 (63.65) | 0.294 |
| Legislation | 76 (14.93) | 75 (14.82) | 151 (14.88) | 0.961 |
| Involvement of religious and community members | 177 (34.77) | 149 (29.45) | 326 (32.12) | 0.069 |
| Education of people on social media | 147 (28.88) | 127 (25.10) | 274 (27.00) | 0.175 |

donate brain samples for research purposes". Hospitals and the internet (online communication platforms) were the main sources of information on brain donation as for blood sample donation. The main reasons for willingness to donate brain by respondents were: advancement of medicine (59.7%), happiness to save a life (52.4%), and prevention of future disease (43.5%). Respondents indicated media publicity (64.4.%) and education (63.7%) as common ways of further promoting brain donation for research as in Table 3.

## Willingness for blood/brain donation/storage and sharing

Table 4 summarizes the willingness for blood/brain donation/sharing and reuse. Majority of participants were willing to give consent for blood sample donation for research for themselves (75.3%) and on behalf of their family members (73.9%).

**Table 4. Willingness towards blood/brain sample donation/sharing/reuse.**

| Variable/question | Male (n = 509) | Female (n = 506) | Total (n = 1015) | p-value |
|---|---|---|---|---|
| | n (%) | n (%) | n (%) | |
| *Willingness to give consent for blood sample donation for genetic research and storage* | | | | |
| Yes | 393 (77.21) | 371 (73.32) | 764 (75.27) | 0.151 |
| No | 116 (22.79) | 135 (26.68) | 251 (24.73) | |
| *Willingness to give consent for a family member blood sample donation for genetic research and storage* | | | | |
| Yes | 371 (72.89) | 379 (74.90) | 750 (73.89) | 0.465 |
| No | 138 (27.11) | 127 (25.10) | 265 (26.11) | |
| *Blood fractions from me can be shared with other researchers* | | | | |
| Yes | 352 (69.16) | 329 (65.02) | 681 (67.09) | 0.161 |
| No | 157 (30.84) | 177 (34.98) | 334 (32.91) | |
| *Brain tissues from me can be shared with other researchers* | | | | |
| Yes | 216 (42.44) | 156 (30.83) | 372 (36.65) | <0.001 |
| No | 293 (57.56) | 350 (69.17) | 643 (63.35) | |
| *Brain images from me can be shared with other researchers* | | | | |
| Yes | 299 (58.74) | 260 (51.38) | 559 (55.07) | 0.018 |
| No | 210 (41.26) | 246 (48.62) | 456 (44.93) | |
| **Questions related to bio-rights** | | | | |
| *Do you think participants in researches should have control on how their biological specimens will be used?* | | | | |
| Yes | 214 (42.04) | 228 (45.06) | 442 (43.55) | 0.551 |
| No | 199 (39.10) | 182 (35.97) | 381 (37.54) | |
| *How much control should/can individuals have regarding how their biological specimens will be used in research?* | | | | |
| None | 206 (40.47) | 206 (40.71) | 412 (40.59) | 0.337 |
| Little | 137 (26.92) | 111 (21.94) | 248 (24.43) | |
| Much | 96 (18.86) | 115 (22.73) | 211 (20.79) | |
| Total | 43 (8.45) | 45 (8.89) | 88 (8.67) | |

Only two out of ten participants were willing to donate brain after death (18.8%, M: 22.6%, F: 15.0%, p = 0.002) as shown in Table 3. Over 6 out of 10 participants were willing to share their blood samples with researchers other than those they initially consent to use their data. However, just about three out of ten were willing to share brain tissues with other researchers beyond those they initial consent for participation with (36.7%, M: 42.2%, F: 30.8%, p<0.001) (Table 4). Males were more willing than females to share their brain images with other researchers (58.7% versus 51.4%, p = 0.018). Whereas about 43% of the respondents wanted to have some degree of control over their biological samples and their usage, only 8.7% wanted total control (Table 4).

## Awareness and perception of participants about informed consent

Responses to questions related to informed consent are detailed in Table 5. Only a third of the participants had heard of informed consent (31.7%; M: 35.9%, F: 27.5%, p = 0.004) with a preference for the broad consent (58.1%).

**Table 5. Awareness and perception about informed consent.**

| Variable/question | Male (n = 509) | Female (n = 506) | Total (n = 1015) | p-value |
|---|---|---|---|---|
| | n (%) | n (%) | n (%) | |
| *Heard of informed consent. (% Yes)* | 183 (35.95) | 139 (27.47) | 322 (31.72) | 0.004 |
| *Types of informed consent preferred* | | | | |
| Broad | 104 (56.83) | 83 (59.71) | 187 (58.07) | 0.447 |
| Restricted | 35 (19.13 | 26 (18.71) | 61 (18.94) | |
| Tiered | 9 (4.92) | 10 (7.19) | 19 (5.90) | |
| Dynamic | 30 (16.39) | 14 (10.07) | 44 (13.66) | |
| *Persons to be involved before giving informed consent* | | | | |
| No one | 84 (45.90) | 43 (30.94) | 127 (39.44) | 0.086 |
| Spouse | 49 (26.78) | 50 (35.97) | 99 (30.75) | |
| Children | 14 (7.65) | 16 (11.51) | 30 (9.32) | |
| Parents | 24 (13.11) | 21 (15.11) | 45 (13.98) | |
| Religious leaders and others | 12 (6.56) | 9 (6.47) | 21 (6.52) | |
| *It is best to use generic informed consent for community* | | | | |
| Agree | 89 (48.63) | 79 (56.83) | 168 (52.17) | 0.037 |
| Disagree | 92 (50.27) | 54 (38.85) | 146 (45.34) | |
| *Perception about informed consent (% agreed)* | | | | |
| Broad informed consent should be used for genomic research. | 244 (47.94) | 260 (51.38) | 504 (49.66) | 0.546 |
| Consent forms should include a separate section relating to storage and future use of samples and data. | 314 (61.69) | 339 (67.00) | 653 (64.33) | 0.147 |
| It is personal choice to give blood for research. | 372 (73.08) | 394 (77.87) | 766 (75.47) | 0.206 |
| Any blood sample collected from me must not be used for any other secondary use. | 220 (43.22) | 251 (49.60) | 471 (46.40) | 0.122 |
| I will participate in genomic research if my community leader agrees | 126 (24.75) | 182 (35.97) | 308 (30.34) | <0.001 |
| Donor must be contacted each time the sample is to be re-used. | 189 (37.13) | 217 (42.89) | 406 (40.00) | 0.109 |
| I feel it's a criminal offence to make profit from sample collected from me. | 268 (52.65) | 289 (57.11) | 557 (54.88) | 0.224 |

Close to half of the participants agreed on a generic informed consent at the community level (52.2%; M: 48.6%, F: 56.8%, p = 0.037). While the majority (74.5%) of the participants agreed that blood donation for research is a personal choice, two-thirds indicated that consent forms should have a separate section on storage and future use of samples and data. Also, a third of the respondents indicated that they would participate in genomic research if their community leaders were involved (30.3%; M: 24.8%, F: 35.9%, p<0.001) (Table 5).

## Association of participant characteristics with willingness to donate share blood/brain sample

Table 6 summarizes the association of participant characteristics with willingness to donate share blood/brain fraction. Participants with tertiary education were more willing to donate brain samples for research [OR: 4.04 (C.I: 1.11–14.76) p = 0.034]; permit sharing their brain tissues with other researchers [OR: 3.82 (C.I: 1.51–9.68) p = 0.005], give consent for blood donation for genetic research [OR: 3.45 (C.I: 1.60–7.42), p<0.01] and share their blood samples with other researchers [OR: 2.59 (C.I: 1.28–5.22), p = 0.002] as compared with participants without any formal education.

The odds of sharing of brain tissues with other researchers was 1.8 fold higher among those aged $\geq$ 50 years as compared to < 50 years [OR: 1.8 (C.I: 1.2–2.8). p = 0.007]. The Ga ethnic groups in Ghana were more willing to give consent for blood donation (OR: 6.6 (C.I: 1.7–24.3) p = 0.005); more willing to permit sharing their blood fraction with other researchers (OR: 5.1 (C.I: 1.7–14.8) p = 0.003); more willing to share their brain fractions with other researchers (OR: 3.7 (C.I: 1.5–9.3) p = 0.006) and also more willing to permit sharing their brain images with other researchers (OR: 3.9 (C.I: 2.1–7.5) p<0.001) as compared with the Yoruba ethnic groups in Nigeria.

## Association of demographic characteristics with awareness about informed consent

The association of demographic characteristics with awareness about informed consent is presented in Table 7. Similar to our findings above, tertiary education was significantly associated with awareness about informed consent [OR: 6.95 (C.I: 2.8–12.3), p<0.001) as compared with those with no formal education Table 7.

Awareness about informed consent was higher among the Ewe groups in Ghana [OR: 5.4 (C.I: 1.4–20.9), p = 0.014] and Igbo groups in Nigeria [OR: 3.1 (C.I: 1.2–7.9), p = 0.017) as compared with the Yoruba groups. Compared to highly skilled groups, all other occupational groups were less likely to be aware of informed consent processes and options.

## Discussion

There is a dearth of research examining concepts within genomic research including informed consent, neurobiobanking and awareness and willingness to donate biological samples such as blood and brain within sSA. Our study findings indicate that participants had lower levels of awareness about brain sample donation for research and low levels of willingness to donate brain samples (20%) as compared to blood samples (75%). These findings are consistent with findings from the IBADAN Brain Bank Project in Nigeria in which the awareness related to brain donation was found to be lower than for other organs [10]. There is generally low level of awareness of biobanking and organ donation globally, and brain donation for research is still an evolving concept in sSA [2, 18]. In our study, approximately a third had heard of blood sample donation for research, whereas only about a tenth had heard of collecting and storing

**Table 6. Association of socio demographic characteristics and willingness to donate/share blood/brain samples.**

| Variable/characteristic | AOR (95% CI) | p-value |
|---|---|---|
| *Willingness towards donation of brain sample for research* | | |
| Gender | | |
| Male | 1.37 (0.95–1.97) | 0.096 |
| Female | 1 | |
| Domicile | | |
| Rural | 0.86 (0.34–2.21) | 0.760 |
| Semi-urban | 0.93 (0.61–1.40) | 0.727 |
| Urban | 1 | |
| Education | | |
| None | 1 | |
| Arabic | 3.01 (0.25–35.66) | 0.382 |
| Primary | 1.65 (0.43–6.34) | 0.470 |
| Secondary | 3.29 (0.95–11.48) | 0.061 |
| Tertiary | 4.04 (1.11–14.76) | 0.034 |
| Religion | | |
| Christianity | 1 | |
| Islam | 1.35 (0.87–2.09) | 0.184 |
| Others | 3.69 (0.31–44.68) | 0.304 |
| Occupation | | |
| Highly skilled/professionals | 1 | |
| Skilled | 1.53 (0.72–3.28) | 0.267 |
| Semi-skilled | 2.04 (0.88–4.74) | 0.097 |
| Manual work | 1.79 (0.68–4.77) | 0.239 |
| Not working | 1.17 (0.48–2.84) | 0.731 |
| Student | 3.79 (1.61–8.94) | 0.002 |
| *Willingness to permit sharing of brain tissues with other researchers* | | |
| Age group | | |
| < 50 | 1 | |
| > = 50 | 1.81 (1.18–2.77) | 0.007 |
| Gender | | |
| Male | 1.31 (0.96–1.79) | 0.093 |
| Female | 1 | |
| Domicile | | |
| Rural | 0.57 (0.25–1.30) | 0.182 |
| Semi-urban | 1.01 (0.69–1.46) | 0.961 |
| Urban | 1 | |
| Education | | |
| None | 1 | |
| Arabic | 0.67(0.07–6.70) | 0.732 |
| Primary | 1.60 (0.61–4.23) | 0.343 |
| Secondary | 2.87 (1.18–6.99) | 0.021 |
| Tertiary | 3.82 (1.51–9.68) | 0.005 |
| Religion | | |
| Christianity | 1 | |
| Islam | 1.34 (0.92–1.97) | 0.130 |
| Others | 1.06 (0.09–11.87) | 0.963 |
| Ethnic group | | |

(*Continued*)

**Table 6.** (Continued)

| Variable/characteristic | AOR (95% CI) | p-value |
|---|---|---|
| Yoruba | 1 | |
| Igbo | 1.37 (0.61–3.11) | 0.448 |
| Hausa | 1.50 (0.85–2.66) | 0.162 |
| Akan | 1.69 (0.72–3.95) | 0.228 |
| Ga/Adangbe | 3.67 (1.45–9.27) | 0.006 |
| Ewe | 3.87 (1.13–13.28) | 0.032 |
| Others | 1.68 (0.96–2.92) | 0.068 |
| Occupation | | |
| Highly skilled/professionals | 1 | |
| Skilled | 1.43 (0.81–2.53) | 0.221 |
| Semi-skilled | 1.88 (0.96–3.67) | 0.066 |
| Manual work | 1.95 (0.94–4.06) | 0.073 |
| Not working | 1.26 (0.65–2.47) | 0.492 |
| Student | 3.86 (1.99–7.48) | <0.001 |
| *Willingness to give consent to blood donation for genetic research* | | |
| Gender | | |
| Male | 1.17 (0.82–1.67) | 0.377 |
| Female | 1 | |
| Education | | |
| None | 1 | |
| Arabic | 2.72 (0.55–13.54) | 0.220 |
| Primary | 2.24 (1.04–4.82) | 0.039 |
| Secondary | 2.08 (1.04–4.18) | 0.040 |
| Tertiary | 3.45 (1.60–7.42) | 0.002 |
| Religion | | |
| Christianity | 1 | |
| Islam | 1.08 (0.73–1.59) | 0.694 |
| Others | 3.16 (0.27–36.25) | 0.356 |
| Ethnic group | | |
| Yoruba | 1 | |
| Igbo | 1.63(0.69–3.88) | 0.266 |
| Hausa | 3.29 (1.70–6.35) | <0.001 |
| Akan | 8.43 (2.78–25.55) | <0.001 |
| Ga/Adangbe | 6.55 (1.77–24.32) | 0.005 |
| Others | 2.93 (1.55–5.53) | 0.001 |
| Occupation | | |
| Highly skilled/professionals | 1 | |
| Skilled | 0.74 (0.39–1.42) | 0.366 |
| Semi-skilled | 1.07 (0.48–2.36) | 0.870 |
| Manual work | 0.80 (0.35–1.83) | 0.604 |
| Not working | 0.49 (0.24–1.02) | 0.057 |
| Student | 1.47 (0.65–3.16) | 0.326 |

brain for research. However, the awareness levels of participants in the IBADAN Brain Bank Project was higher as compared to ours [10]. This could be attributed to the higher age group of participants in the IBADAN Brain Bank Project (mean age 46.3 years), though these differences may also stem from different cultural attitudes towards donation, religious beliefs or low

**Table 7. Association of socio demographic characteristics and awareness of consent process.**

| Characteristics | AOR (95% CI) | p-value |
|---|---|---|
| Age group | | |
| < 50 | 1 | |
| > = 50 | 1.28 (0.83–1.98) | 0.266 |
| Gender | | |
| Male | 1.26 (0.90–1.75) | 0.173 |
| Female | | |
| Domicile | | |
| Rural | 1.93 (0.92–4.03) | 0.080 |
| Semi-urban | 1.24 (0.84–1.83) | 0.227 |
| Urban | 1 | |
| Education | | |
| None | 1 | |
| Arabic | 0.95 (0.09–9.31) | 0.965 |
| Primary | 1.08 (0.42–2.83) | 0.870 |
| Secondary | 2.04 (0.86–4.85) | 0.107 |
| Tertiary | 6.95 (2.79–12.28) | <0.001 |
| Marital status | | |
| Single | 1 | |
| Married | 2.44 (1.13–5.27) | 0.023 |
| Formerly married | 1.59 (0.65–3.89) | 0.313 |
| Religion | | |
| Christianity | 1 | |
| Islam | 1.01 (0.68–1.52) | 0.945 |
| Others | 3.37 (0.35–32.18) | 0.293 |
| Ethnic group | | |
| Yoruba | 1 | |
| Igbo | 3.09 (1.23–7.78) | 0.017 |
| Hausa | 1.07 (0.53–2.15) | 0.852 |
| Akan | 0.89 (0.37–2.21) | 0.818 |
| Ga/Adangbe | 1.83 (0.68–4.91) | 0.232 |
| Ewe | 5.42 (1.40–20.92) | 0.014 |
| Others | 2.04 (1.08–3.89) | 0.029 |
| Occupation | | |
| Highly skilled/professionals | 1 | |
| Skilled | 0.52 (0.29–0.93) | 0.029 |
| Semi-skilled | 0.36 (0.18–0.73) | 0.004 |
| Manual work | 0.34 (0.16–0.73) | 0.001 |
| Not working | 0.45 (0.23–0.89) | 0.027 |
| Student | 0.18 (0.09–0.37) | <0.001 |

levels of trust in public institutions (which may result from previous breaches of trust) as highlighted by Tindana et al. (2012) [19]. Some studies have indicated that biobanking knowledge, type of donated tissue, research purpose, concerns over the safety of the data, preferred type of consent, and trust towards biobanks, affect willingness to donate [6]. Indeed, over half of the participants in our study indicated that they were not willing to donate brain because they wanted to go back to God complete (religious beliefs). Studies conducted in Scandinavian countries (such as Sweden and Finland) highlight the positive correlation between knowledge

and positive opinions on biobanks with respondents' willingness to donate; where the knowledge about biobanks is highest, 83% of Finns and 86% of Swedes declared such willingness [20–22]. Awareness about biobanking is generally low globally and not only confined to the African Region. In the 2010 Eurobarometer study, for instance, two-thirds of Europeans have never heard about biobanks and less than 2% search for information about biobanking [13]. Low levels of awareness on biobanking and the increased willingness to donate (mainly blood and not brain) in our study calls for stakeholders input (general public, religious leaders, scientists, industry, and non-governmental organizations) through community-based participatory research and citizen science approaches to identify research priorities and actively involve sample donors in biobanking process and guidelines to further scientific advancements [8, 23, 24].

An important concept in genetic research and biobanking is the process of informed consent. This is required for several reasons including storage of samples (sometimes for an indefinite period) and to use samples for unspecified future research. Only a third of our study participants were aware of informed consent. This could be attributed to the profile of the participants who were largely laypersons in the studied communities with no prior information on or engagement with biomedical research. Nevertheless, participants generally had a preference for the broad type of consent (a process by which individuals donate their samples for a broad range of unspecified future studies with some restriction) [5, 25, 26]. Although, the broad type of consent has been proposed as an appropriate consent model in African genomics research and biobanking [27], it has been linked with the risk of exploitation of African research populations [4, 25, 28]. Nevertheless, this approach reduces the financial and logistical barriers to researchers, and the burden to participants, which may be a particular challenge in many African research settings [8]. Recommendations for the use of the broad consent models has been to include governance mechanisms that incentivize biobanks to promote the interests of biological sample donors as well as communities' health and research needs [8]. Indeed, it was observed in our study that over half of the participants indicated they agreed with a generic consent at the community level and a third of the participants indicated that they were more likely to participate in genomic studies if their community leaders agree. However, over half of the participants did not know of any guidelines regulating blood and brain biobanking. The role of a community engagement approach has been found to be a critical component in the ethical conduct of health research and is particularly pertinent in communitarian societies such as Africa [29]. Also, national guidelines at the country level within Africa are important for biobanking to eliminate what has been described as exploitative "parachute" research (a practice whereby scientists in high-income countries go to low-income countries to collect specimens and publish findings in prestigious journals without properly crediting collaborators in Low-and-Middle Income Countries or providing tangible benefits to study communities) [8, 24].

Other factors, such as socio-demographic characteristics, were also associated with willingness to participate in biobanking research. Educational attainment (tertiary education) and male gender were found to be important predictors of willingness to donate and share both blood and brain samples for research and having prior awareness about informed consent. While religious beliefs did not seem to influence participants' willingness to sample donation and sharing in our study, in a British study, non-believers and less religious persons were more interested in donation [18]. Although religious beliefs may not be a determining factor in biobank participation, it is expected to provide comfort into the willingness to enroll in research initiatives. For instance, the linkage of Islam with scientific knowledge and advancement, may be influential in increasing awareness towards genetics and biobanking [30]. In developing interventions, it is thus important to promote better representation of

socioeconomic diversity including religion in research leadership and ensure tailored health education materials of appropriate literacy to expand genetic education for increased public awareness and understanding. By expanding participation rates among the diverse populations within Africa, opportunities exist to better understand the genomic diversity representative across the continent [27, 28, 31]. Our study also highlights the influence of certain ethnic groups (Ewe and Igbo groups), who were likely to be aware of informed consent as compared to the others (Yoruba). This finding needs further studies to explain this observation as it is beyond the scope of this current study.

This study contributes further to our understanding of communities in Ghana and Nigeria on views and attitudes towards biological sample donation/sharing and informed consent. It also reinforces the importance of involving the public in a more transparent dialogue about the use of biological samples to encourage greater public involvement and support for this area given the low awareness levels. It indicates the need for good governance concerning biological samples and their associated data, which requires complex discussions around community engagement, public learning and understanding of science and ethical principles of informed consent. The findings of our study should be considered in light of the following limitations: first, it relies on self-reported data and not objective measurements of awareness, attitudes and biobanking knowledge, which could be influenced by a social desirability bias. Second, as no validated tools exist, the authors relied on methods that have been utilized in other genetics literature. Nevertheless, our study's large sample size, diverse coverage, and rigorous sampling strategy of participants are potential strengths.

## Conclusion

Our study findings demonstrate that despite inadequate awareness of biobanking, sample donation and informed consent, there is a high level of public support for, and willingness to contribute to biobanking related to blood donation (than brain). Individuals with higher educational levels are more willing to donate samples indicating the need to inform a broader public including the older generation and people in rural areas about the role of research biobanks. Improved public education through strategies including the social media; communication with representatives of patients' organizations, local community and other stakeholders; promotion of active participation and engagement of the community/donors in promoting the idea of biobanking while taking care of the cultural and religious diversities of the donors are recommended to mitigate some of the concerns.

## Supporting information

**S1 File. Questionnaire_survey.**
(PDF)

## Author Contributions

**Conceptualization:** Oyedunni Arulogun, Rufus Akinyemi.

**Data curation:** Joshua Akinyemi.

**Formal analysis:** Joshua Akinyemi.

**Funding acquisition:** Oyedunni Arulogun, Rufus Akinyemi.

**Project administration:** Shadrack Asibey.

**Supervision:** Arti Singh.

**Writing – original draft:** Arti Singh.

**Writing – review & editing:** Oyedunni Arulogun, Michelle Nichols, Benedict Calys-Tagoe, Babatunde Ojebuyi, Carolyn Jenkins, Reginald Obiako, Albert Akpalu, Fred Sarfo, Kolawole Wahab, Adeniyi Sunday, Lukman F. Owolabi, Muyiwa Adigun, Ibukun Afolami, Olorunyomi Olorunsogbon, Mayowa Ogunronbi, Ezinne Sylvia Melikam, Ruth Laryea, Shadrack Asibey, Wisdom Oguike, Lois Melikam, Abdullateef Sule, Musibau A. Titiloye, Isah Suleiman Yahaya, Abiodun Bello, Rajesh N. Kalaria, Ayodele Jegede, Mayowa Owolabi, Bruce Ovbiagele, Rufus Akinyemi.

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
