## [Decision Letter · Decision Letter 0]

31 Jan 2022

PONE-D-21-25570Biological sample donation and informed consent for neurobiobanking: Evidence from a community survey in Ghana and Nigeria.PLOS ONE

Dear Dr. Singh,

Thank you for submitting your manuscript to PLOS ONE. After careful review, we consider it to have merit but does not fully meet PLOS ONE’s publication criteria as it currently stands. Therefore, we invite you to submit a revised version of the manuscript that addresses the points raised during the review process.

Please respond to all the comments by the three reviewers. In addition, please provide in the Supplement copies of the questionnaire and its English translation as well a file with the raw data.==============================

We look forward to receiving your revised manuscript.

Kind regards,

Helena Kuivaniemi, MD, PhD

Academic Editor

PLOS ONE

https://journals.plos.org/plosone/s/file?id=ba62/PLOSOne_formatting_sample_title_authors_affiliations.pdf”

“The study and investigators are funded by the National Institute of Health grants SIREN (U54HG007479), SIBS Genomics (R01NS107900), SIBS Gen Gen (R01NS107900 – 02S1), ARISES  (R01NS115944-01), and CVD Supplement (3U24HG009780-03S5).”

“RA National Institute of Health grants SIREN (U54HG007479), SIBS Genomics (R01NS107900), SIBS Gen Gen (R01NS107900 – 02S1), ARISES  (R01NS115944-01), and CVD Supplement (3U24HG009780-03S5).

Additional Editor Comments:

Please provide these items as supplementary material:

1) The original questionnaire used in the study

2) English translation of the questionnaire

3) Raw data collected in the study. E.g., csv files that includes the responses of all >1,000 participants to all the questions so that others can re-analyze the data later.

Reviewers' comments:

Reviewer's Responses to Questions

**Comments to the Author**

1. Is the manuscript technically sound, and do the data support the conclusions?

Reviewer #1: Yes

Reviewer #2: Yes

Reviewer #3: Yes

2. Has the statistical analysis been performed appropriately and rigorously? 

Reviewer #1: Yes

Reviewer #2: Yes

Reviewer #3: Yes

3. Have the authors made all data underlying the findings in their manuscript fully available?

Reviewer #1: Yes

Reviewer #2: Yes

Reviewer #3: No

4. Is the manuscript presented in an intelligible fashion and written in standard English?

Reviewer #1: Yes

Reviewer #2: Yes

Reviewer #3: Yes

5. Review Comments to the Author

Reviewer #1: Thank you for this interesting and important manuscript. This report of awareness and understanding of basic elements pertaining to biobanking among African community participants is an important step toward establishing these necessary research resources in Africa. I applaud and encourage these efforts. This manuscript is generally well-written and pertinent to the scientific community, and I recommend it for publication pending minor revisions. The manuscript would benefit from a careful review for grammatical and punctuation errors. Please see attached file for detailed comments to the authors.

Reviewer #2: Thank you for a well written and conceptualised study. In addition, I thank the authors for performing this much needed study and believe that it will be used by many researchers in the future. I have no major comments but do have a few minor comments that the authors should address:

1. From line 154 onwards where the questionnaire is described: this section is hard to read and follow. My suggestions would be to either provide a paragraph based summary of the questions and include the questionnaire in the supplementary. Any other ways to improve the follow would also be acceptable but I would include a copy of the questionnaire in the supplementary regardless.

2. Please define all abbreviations used in the text and tables e.g. "Ref" in Table 1

3. Please ensure that when quoting a percentage, the presence/lack of space between the number and the % is consistent

4. Please ensure that the use of decimals (and the number of decimals) is kept consistent throughout the manuscript

5. With regards to the question "Willingness to donate brain after death", was this question subdivided into just a sample of brain and then another regarding complete brain donation? I suspect that more individuals would have been willing to donate a sample of their brain than their complete brain.

6. Reformatting of all tables is needed, particularly those reporting questionnaire results, it is difficult to follow as it. All tables should be reviewed for language and grammar errors e.g. Table 3: "I don't just want to" and "Actions for prmoting..."

7. Please rephrase line 247, 89% is not 8 out of 10 respondents

8. Line 254: 80%" (remove the ")

9. Inclusion of "however" on line 269 should be removed.

10. Please rephrase the sentence (and potentially create shorter sentence) from lines 270 till 274.

11 Table 5: please provide the possible answers (I'm assuming agree/disagree) to the "Perception about informed consent" section

12. Please rethink how the sentence on line 307 starts with "Age > 50 years..."

13. Line 357: "concept" should be consent

14. Please replace all occurence of "just about" with "approximately"

15. Line 457: Change to "Our study should be considered in light of the following limitations"

16. Please rephrase how line 464 starts, the inclusion of "established" here is not appropriate.

Reviewer #3: This is an interesting manuscript, which assesses the willingness of community members in Ghana and Nigeria to donate biological samples (brain and blood). The authors performed the analysis in a large sample size. The results are important and take into account ethnic groups as well, which, although reducing sample size somewhat, also yielded interesting results. The findings can indeed be utilised to guide future engagement activities as far as research with biological samples is involved, in Ghana and Nigeria, although the authors do not attempt to extrapolate the data to other African countries (which I understand may be difficult). Overall, a study which will be of much interest to African researchers, although I am not sure how much of a global reach the findings will have. The article requires a thorough proofread. There are many grammatical and formatting errors that should be corrected: e.g. Table 7 needs to be reformatted as it is impossible to read the values in the last column; "sSA" is used as an abbreviation only in the Discussion, and is not defined, even though "sub-Saharan Africa" is used in the Introduction; the Discussion section needs to be divided into paragraphs.

6. PLOS authors have the option to publish the peer review history of their article (what does this mean?). If published, this will include your full peer review and any attached files.

Reviewer #1: No

Reviewer #2: No

Reviewer #3: No

---

## [Author Response · Author response to Decision Letter 0]

16 Mar 2022

We have included a response letter (attached) with a response to each of the questions raised by the reviewers.

---

## [Decision Letter · Decision Letter 1]

31 Mar 2022

PONE-D-21-25570R1Biological sample donation and informed consent for neurobiobanking: Evidence from a community survey in Ghana and Nigeria.PLOS ONE

Dear Dr. Singh,

Thank you for submitting your manuscript to PLOS ONE. After careful evaluation, we consider it to have merit but the manuscript does not fully meet PLOS ONE’s publication criteria as it currently stands. Therefore, we invite you to submit a revised version of the manuscript that addresses the points raised during the review process.

Please pay special attention to formatting irregularities and grammatical errors.

We look forward to receiving your revised manuscript.

Kind regards,

Helena Kuivaniemi, MD, PhD

Academic Editor

PLOS ONE

Journal Requirements:

Reviewers' comments:

Reviewer's Responses to Questions

**Comments to the Author**

1. If the authors have adequately addressed your comments raised in a previous round of review and you feel that this manuscript is now acceptable for publication, you may indicate that here to bypass the “Comments to the Author” section, enter your conflict of interest statement in the “Confidential to Editor” section, and submit your "Accept" recommendation.

Reviewer #1: (No Response)

Reviewer #2: All comments have been addressed

2. Is the manuscript technically sound, and do the data support the conclusions?

Reviewer #1: Yes

Reviewer #2: Yes

3. Has the statistical analysis been performed appropriately and rigorously? 

Reviewer #1: Yes

Reviewer #2: Yes

4. Have the authors made all data underlying the findings in their manuscript fully available?

Reviewer #1: Yes

Reviewer #2: Yes

5. Is the manuscript presented in an intelligible fashion and written in standard English?

Reviewer #1: No

Reviewer #2: Yes

6. Review Comments to the Author

Reviewer #1: The edits made by the authors have significantly improved this manuscript and I am very supportive of publication. My only remaining comments pertain to the persistent formatting irregularities and grammatical errors throughout the manuscript. While this issue has been much improved, there remain several errors that ideally should be corrected prior to publication. I have outlined some examples below but this is not an exhaustive list.

Introduction:

Inconsistencies between “African sub - region” and “African sub region”

Methods:

Line 172 - “1.Demographics” is italicized as though it is a subheader but none of the other items in the numbered list are.

Lines 183-186 - grammar/punctuation problems

Results:

The table format is improved but spacing and formatting issues remain, including random spaces, inconsistent capitalization, and “%” not be used in all tables.

Table 5 displays the questions differently from previous tables

Tables 6 & 7 formats are difficult to read without more helpful visual formatting and should have similar format.

Lines 236-238 - grammar/punctuation problems

Line 258 - “blood/bank” should be “blood/brain”

Line 276 - “as” should be “are”

Discussion

Line 376 - there is a redundant comma

Line 378 - there is a redundant word “levels”

Reviewer #2: (No Response)

7. PLOS authors have the option to publish the peer review history of their article (what does this mean?). If published, this will include your full peer review and any attached files.

Reviewer #1: No

Reviewer #2: No

---

## [Author Response · Author response to Decision Letter 1]

8 Apr 2022

Dear Sir/Madam, 

We have responded to the minor comments raised by reviewer 1 and attached a letter with our responses. We have also included a tracked copy of the changes requested. 

Thanking you

---

## [Editor Report · Decision Letter 2]

14 Apr 2022

Biological sample donation and informed consent for neurobiobanking: Evidence from a community survey in Ghana and Nigeria.

PONE-D-21-25570R2

Dear Dr. Singh,

We’re pleased to inform you that your manuscript has been judged scientifically suitable for publication and will be formally accepted for publication once it meets all outstanding technical requirements.

Congratulations!

Kind regards,

Helena Kuivaniemi, MD, PhD

Academic Editor

PLOS ONE
---

## [Editor Report · Acceptance letter]

3 Aug 2022

PONE-D-21-25570R2 

Biological sample donation and informed consent for neurobiobanking: Evidence from a community survey in Ghana and Nigeria. 

Dear Dr. Singh:

I'm pleased to inform you that your manuscript has been deemed suitable for publication in PLOS ONE. Congratulations! Your manuscript is now with our production department. 

Kind regards, 

on behalf of

Professor Helena Kuivaniemi 

Academic Editor

PLOS ONE